# EPH/Ephrin Signaling in Normal Hematopoiesis and Hematologic Malignancies: Deciphering Their Intricate Role and Unraveling Possible New Therapeutic Targets

**DOI:** 10.3390/cancers15153963

**Published:** 2023-08-04

**Authors:** Ioanna E. Stergiou, Stavros P. Papadakos, Anna Karyda, Ourania E. Tsitsilonis, Meletios-Athanasios Dimopoulos, Stamatios Theocharis

**Affiliations:** 1Department of Pathophysiology, School of Medicine, National and Kapodistrian University of Athens, 11527 Athens, Greece; stergiouioa@med.uoa.gr; 2First Department of Pathology, School of Medicine, National and Kapodistrian University of Athens, 11527 Athens, Greece; stpap@med.uoa.gr (S.P.P.); karyanna@med.uoa.gr (A.K.); 3Flow Cytometry Unit, Department of Biology, School of Science, National and Kapodistrian University of Athens, 15784 Athens, Greece; rtsitsil@biol.uoa.gr; 4Department of Clinical Therapeutics, School of Medicine, National and Kapodistrian University of Athens, Alexandra Hospital, 11528 Athens, Greece

**Keywords:** EPHs, ephrins, hematopoiesis, hematologic malignancies, therapeutic targeting

## Abstract

**Simple Summary:**

The EPH/ephrin signaling axis holds a pivotal role in a plethora of physiologic processes during embryonic development and adult life, while more recent studies highlight its implication in the pathogenesis of multiple types of solid tumors. Different patterns of EPH/ephrin expression have been identified in normal hematopoiesis concerning the maintenance, differentiation, and function of hematopoietic stem cells and their mature offspring. Emerging data suggest that EPH/ephrin aberrations contribute to the pathogenesis of various hematologic malignancies, exerting both tumor-promoting and tumor-suppressive functions. Thus, unraveling the intricate role of EPH/ephrin signaling in the setting of hematologic neoplasia can contribute to the development of promising therapeutic targets to overcome unmet needs.

**Abstract:**

Erythropoietin-producing hepatocellular carcinoma receptors (EPHs) represent the largest family of receptor tyrosine kinases (RTKs). EPH interaction with ephrins, their membrane-bound ligands, holds a pivotal role in embryonic development, while, though less active, it is also implicated in various physiological functions during adult life. In normal hematopoiesis, different patterns of EPH/ephrin expression have been correlated with hematopoietic stem cell (HSC) maintenance and lineage-committed hematopoietic progenitor cell (HPC) differentiation, as well as with the functional properties of their mature offspring. Research in the field of hematologic malignancies has unveiled a rather complex involvement of the EPH/ephrinsignaling pathway in the pathophysiology of these neoplasms. Aberrations in genetic, epigenetic, and protein levels have been identified as possible players implicated both in tumor progression and suppression, while correlations have also been highlighted regarding prognosis and response to treatment. Initial efforts to therapeutically target the EPH/ephrin axis have been undertaken in the setting of hematologic neoplasia but are mainly confined to the preclinical level. To this end, deciphering the complexity of this signaling pathway both in normal and malignant hematopoiesis is necessary.

## 1. Introduction

Erythropoietin-producing hepatocellular carcinoma receptors (EPHs) have been identified since the late 1980s and represent the largest family of receptor tyrosine kinases (RTKs) [1]. Ephrins are membrane-bound ligands recognized by EPHs [2]. Through the delivery of either attractive or repulsive signals that guide cell movement, the EPH/ephrin signaling pathway is critical for embryonic development [3], orchestrating cell migration and adhesion during gastrulation, somitogenesis, and the establishment of tissue and organ boundaries [4]. The developing nervous system presents the highest expression levels of EPHs and ephrins, whose interactions direct axon formation [5]. Moreover, during angiogenesis, EPH/ephrin signaling, driven by the specific pattern of distribution of EPHs and ephrins in vasculature, defines the vascular arterial and venous borders [6,7,8]. Though EPHs exert more prominent functions during embryonic development and are generally less active in adult tissues, they continue to be implicated in organ maintenance and regeneration, synaptic plasticity, nerve and vascular remodeling, and response to injury [9]. They are also associated with stem cell maintenance, migration, differentiation, and proliferation [9,10]. Additionally, EPH/ephrin expression in both innate and adaptive immune cells affects their development, trafficking, and activation [11,12].

The EPH/ephrin signaling pathway has also been implicated in disease. Its emerging roles have more recently been studied in non-malignant diseases such as atherosclerosis and fibrosis [13,14,15]. In neoplasia, overexpression of EPHs and ephrins has been identified, especially in association with undifferentiated or progenitor cells and in correlation with tumor development, proliferation, and vasculature (reviewed in [16]). Therefore, the expression of EPHs in malignancies has gained particular interest regarding the pathogenesis, invasion, and metastasis of various neoplasms, including lung, prostate, colon, pancreatic, ovarian, thyroid, tongue, and hepatocellular carcinomas, as well as gliomas and melanomas, while efforts for therapeutic targeting are also under investigation [17,18,19,20,21,22,23,24,25,26,27,28]. Interestingly, depending on the context, EPH/ephrin signaling can exert either tumor-promoting or tumor-suppressive functions [29,30,31,32,33,34,35,36,37,38].

Despite quite extensive research regarding the role of EPH/ephrin signaling in solid tumor biology, reports regarding hematologic malignancies are more limited. In this review, after summarizing the role of EPH/ephrin signaling in normal hematopoiesis, we will focus on its implications in the pathophysiology of various hematologic malignancies and explore its potential therapeutic targeting in the setting of these neoplasms.

## 2. Ephrins and EPHs: Structure and Signaling Pathways

In mammals, fourteen members of the EPH family have been so far identified, and based on their sequence homology, ligand specificity, and structural features, they are subcategorized as EPHA (i.e., EPHA1–8 and EPHA10) and EPHB (i.e., EPHB1-4 and EPHB6) [39,40]. In general, EPHA group members demonstrate preferential ligation to glycosylphosphatidylinositol (GPI)-anchored type A ephrins, while EPHB group members prefer transmembrane type B ephrins [2]. Nine EPHAs binding five ephrinA ligands and five EPHBs binding three ephrinB ligands have been identified in humans [40,41]. EPHs share a similar structure consisting of (a) an immunoglobulin-like ligand-binding extracellular globular domain, (b) a Cys-rich domain, which includes sushi and epidermal growth factor (EGF)-like motifs, (c) fibronectin domains, (d) a transmembrane domain, and (e) a tyrosine kinase intracellular domain next to a sterile alpha motif (SAM) and a post-synaptic density protein (PSD95), Drosophila disc large tumor suppressor (Dlg1), and zonula occludens-1 protein (PDZ) domain [1,42]. Type A ephrins present a receptor-binding globular domain next to a GPI-linker, while type B ephrins comprise a receptor-binding globular domain, a transmembrane domain, intracellular phosphorylation, and PDZ domains (Figure 1A) [43]. The binding affinity of each EPH to a certain ephrin varies. For example, for EPHA1, the binding affinity to ephrinA1 is high, to ephrinA3 is lower, and to ephrinA4 is zero [44], while EPHA3 binds with higher affinity to ephrinA5 and EPHB4 to ephrinB2 compared to other ephrins [45,46]. There are also some exceptions to the general “rule” that type A ephrins bind to EPHAs and type B ephrins bind to EPHBs. Namely, type B ephrins can bind to EPHA4 [2], and ephrinA5 can bind to EPHB2 [47].

EPH activation and signaling demonstrate unique features. Given that EPHs bind membrane-bound ephrins, cell-cell contact is necessary for receptor activation. It has been demonstrated that some type A ephrins can be released from the cell surface and are functionally active as soluble molecules [48,49]. These findings propose an alternative signaling mechanism that does not require cell-to-cell contact. Unlike most RTKs, for which dimerization is a prerequisite for activation, EPH activation depends on multimerization, with the assembly of clusters comprising ephrin and EPH molecules intercalating with one another [50,51]. The size of these EPH/ephrin clusters defines the type and intensity of signaling [52,53]. Upon binding of ephrins to EPHs, signaling can be induced in a bidirectional mode. Ephrin binding to EPHs leads to receptor signaling, or “forward signaling”, while EPHs can elicit ephrin signaling, or “reverse signaling” [54,55]. The autophosphorylation of EPH tyrosine residues proximal to the cell membrane is a prerequisite for forward signaling [56]. Typically, type B ephrins’ signaling can take place in the “reverse” direction, being either phosphorylation-dependent or independent. Given the absence of ephrins’ intrinsic catalytic activity, the phosphorylation-dependent reverse signaling initiated by type B ephrins is mediated by the phosphorylation of their intracellular domain by Src kinases [54,57,58]. For phosphorylation-independent reverse signaling, type B ephrins associate with partner proteins [59]. Of note, type A ephrins may also have the potency of “reverse signaling” [60].

“Forward signaling” follows the mode of typical RTK signaling. Ligand binding leads to the activation of the receptor kinase domain, modulating networks of downstream adaptor and effector proteins [61]. Of particular importance amongst them are the Rho and Ras family GTPases and Akt/mammalian target of rapamycin complex 1 (mTORC1) (reviewed in [62]). EPH-mediated regulation of Rho GTPases, such as RhoA, Rac1, and Cdc42, controls cell shape, adhesion, and motility via its effects on the actin cytoskeleton [10,55]. EPH “forward signaling” usually has as a consequence the inhibition of the Ras/extracellular signal-regulated kinase (ERK) pathway [10,55], having variable effects on cell division, differentiation, motility, and gene expression [63,64,65,66]. Of note, there are cases in which the Ras/ERK pathway may be activated by EPH’s “forward signaling” [67,68,69,70]. While RTKs typically activate Akt via the phosphoinositide 3 kinase (PI3K) [71], EPH’s “forward signaling” can either activate [72,73] or suppress Akt [74,75,76,77] (Figure 1B). “Reverse signaling” induced by type B ephrins can activate the Janus kinase (Jak)2/signal transducer and activator of transcription (STAT)3 [78] and Jak1/STAT1/c-Jun N-terminal kinase (JNK)3 [79] pathways, while it can also interact with Grb4-Pak1-Dock180 [80,81] and Par polarity complexes [82] (Figure 1B).

## 3. EPH/Ephrin Expression in Hematopoiesis

During early vertebrate development, both primitive hematopoietic stem cells (HSCs) and endothelial progenitors originate from a common bipotential progenitor cell, the hemangioblast [83]. EPHB4 and ephrinB2 are co-expressed in the yolk sac, the first site of hematopoiesis and vascular development during embryogenesis [84]. Despite the well-established role of EPH/ephrin signaling in angiogenesis [85], many aspects of its implication in hematopoiesis are under investigation. Table 1 summarizes the expression of ephrins and EPHs in normal hematopoietic cells.

### 3.1. Hematopoietic Stem and Progenitor Cells

A plethora of cell-cell and cell-extracellular matrix (ECM) interactions regulate the organization and function of the HSC niche in the bone marrow (BM) microenvironment. EPH/ephrin signaling can modify this fine-tuned “cross-talk”. In the 1990s, the expression of EPHB4 was identified in human HSCs, and it was also associated with myeloid differentiation [100,101]. Since then, a heterogeneous expression of EPHs and ephrins has been documented within the HSC compartment. Using specialized cDNA arrays, Steidl et al. showed that human CD34+ HSCs and hematopoietic progenitor cells (HPCs) express EPHA1 and its ligands ephrinA3 and ephrinA4, thus unveiling a possible role of EPH/ephrin interactions in HSC/HPC localization and function [87]. Ivanova et al. identified ephrinB2 among a group of long-term HSC-specific ligands, probably involved in signaling between HSCs and their microenvironment [86]. The possible functional roles of EPHs and ephrins include the regulation of HSC adhesion, migration, and differentiation, but they have not yet been fully elucidated.

The BM stromal microenvironment contributes to the maintenance and regulation of HSCs and their more mature offspring [102,103]. EPH/ephrin communication contributes to stromal cell-HSC interactions [104], particularly in the context of maintenance and homing of the latter. Nguyen et al. showed that human hematopoietic stem/progenitor cells (HSPCs), but not lineage-committed hematopoietic cells, express EPHA5 and EPHA7, while primary human BM stromal cells express ephrinA5 (the high-affinity binding partner of EPHA5 and EPHA7). EphrinA5 stimulated EPHA5 and/or EPHA7 forward signaling, which in turn promoted HSPC colony formation while also favoring HSPC adhesion and migration. In this case, it has been shown that these effects of ephrinA5 are mediated via activation of Rac1 signaling, and its targeting molecule WAVE [105]. Furthermore, it has been reported that EPHB4 expression by stromal cells plays a role in the regulation of the size of the HSPC pool. Transgenic mice overexpressing EPHB4 demonstrated an expansion of their BM HSPC pool that correlated with a higher frequency of long-term culture-initiating cells (LTC-IC) compared to wild-type (WT) mice. The feeder stromal cells derived from the EPHB4 transgenic mice displayed a greater capacity to support LTC-IC in vitro, while blocking the EPHB4/ephrinB2 interactions resulted in decreased LTC-IC output. Moreover, higher levels of the hematopoietic supportive factors C-X-C motif ligand 12 (CXCL12), angiopoietin 1 (Ang1), interleukin (IL)-6, Fms-like tyrosine kinase 3 (FLT3) ligand (FLT3L), and osteopontin (OP) were expressed by the EPHB4 transgenic mice’s stromal cells compared to WT controls [106]. Similar results were demonstrated in human BM stromal cells, where EPHB4 knockdown led to reduced stromal cell ability to support high-ephrinB2-expressing CD34+ HSC in LTC-IC cultures [106]. Arthur et al. highlighted that ephrinB1 expressed by stromal cells contributes to the support of HSPC function and maintenance. In a mouse model, the authors demonstrated that ephrinB1 loss in osteogenic cells limited their capacity to support HSPC maintenance. EphrinB1-Fc stimulation of human CD34+ HSPCs resulted in EPHB2 activation, enhancing myeloid/erythroid colony formation, whereas functional blocking of either EPHB1 or EPHB2 inhibited the maintenance of LTC-ICs. Both in mouse and human stromal cells, ephrinB1 reverse signaling was found to be necessary for the activation of CXCL12, known to be a critical regulator of HSPC function [107].

EPH/ephrin interactions are also implicated in HSPC adhesion to the BM microenvironment, facilitating their cross-talk not only with stromal cells but also with endothelial cells and the ECM. EPHB4 expression by stromal cells can modulate the expression of ephrinB2 by HPCs. Co-culture of HPCs with different stromal cell lines showed that stromal cell lines expressing EPHB4 could induce the expression of ephrinB2 on HPCs, which in turn favors the adhesion of HPCs to stromal cells. Increased adhesion and decreased transmigration of hematopoietic cells can be the result of ectopic ephrinB2 expression in vitro [97]. Signaling interactions between EPHB4 expressed in BM sinusoids and ephrinB2 expressed in hematopoietic cells control HSPC mobilization from the BM. In mice, blocking EPHB4-ephrinB2 interactions resulted in reduced mobilization of HSPCs and, to a lesser extent, of other committed myeloid cells to the circulation [108]. In a mouse BM transplantation model, ephrinA5 reverse signaling activation via EPHA3 was shown to affect HSC adhesion through an integrin-mediated mechanism. Blockade of the endogenous EPHA-ephrinA interactions by the antagonist EPHA3-Fc resulted in reduced homing of donor stem cells in the BM and their accumulation in the animals’ spleen and peripheral blood (PB) [90]. Lazarova et al. investigated the expression of ephrins in CD34+ and CD133+ cells collected from PB after mobilization. The authors showed that all CD34+ cells and the majority of CD133+ cells expressed EPHA2, while all CD133+ cells and half of the CD34+ cells expressed EPHB2. These findings are suggestive of the EPH/ephrin regulatory role not only in HSCs’ adhesion and migration but also in their differentiation [89].

### 3.2. Erythropoiesis

Apart from its role in the maintenance of the HSC BM niche, the EPH/ephrin axis is also implicated in the differentiation process of HSCs. This function is better described for the erythroid cell lineage and is mainly driven by EPHB4 signaling. Inada et al. demonstrated that about 5% of human BM mononuclear cells were positive for EPHB4 expression. This EPHB4+ population was also c-Kit+, CD34 (low), and glycophorin A (-/low), while further assays showed it consisted exclusively of erythroid progenitors. The proliferation of the majority of these EPHB4+ erythroid progenitors was stem cell factor (SCF)-dependent, identifying them as erythroid burst-forming units (BFU-E). Of note, EPHB4 expression in cord blood cells was upregulated after culture in the presence of SCF [109]. EPHB4 expression was proposed to be restricted to the early stages of erythropoiesis, given that it was initially detected at the immature BFU-E stage, peaked at the late BFU-E stage, waned during the colony-forming unit erythroid (CFU-E) stage, and was undetectable in the early erythroblast stage [110]. Data by Wang et al. demonstrated that transduction of cord blood CD34+ cells with a retroviral vector expressing EPHB4 induced both erythroid and megakaryocytic differentiation and accelerated the transition from a stem cell to a lineage-restricted progenitor phenotype [111].

Expression of ephrinB2, the ligand of EPHB4, is restricted to human BM stromal cells and absent in BM mononuclear cells [110]. During erythroid differentiation of cord blood CD34+ HSCs, EPHB4-ephrinB2 interactions promoted EPHB4 forward signaling. Stromal ephrinB2 possibly contributes to the regulation of erythropoiesis, given that when EPHB4+ HSPCs were co-cultured with ephrinB2-expressing stromal cells, the former detached from the latter and differentiated into mature erythroid cells. It has been proposed that this detachment can lead to increased exposure of HSPCs to soluble cytokines, such as erythropoietin, resulting in their differentiation into mature erythroid cells [96].

More recently, EPHB1 has also been implicated in erythropoiesis. It has been demonstrated that clustering and activation of EPHB1 in developing erythroblasts could be induced by the proteoglycan agrin, leading to the activation of α5β1 integrins. Agrin knockout mice displayed severe anemia due to defective adhesion to macrophages and impaired maturation of erythroid cells [112].

### 3.3. Lymphopoiesis

EPH/ephrin expression has also been studied in the setting of lymphoid development. Aasheim et al. showed that both fetal and adult B cells at all stages of maturation, from early pro-B cells to mature B cells, constitutively express the EPHA4 transcript, with high levels of expression identified in PB B cells. In contrast, EPHA7 transcript expression seems to be developmentally regulated since it was exclusively detected in fetal BM pro-B cells and pre-B cells but not in more mature fetal B-lineage cells, while no expression was detected in adult B cells regardless of maturation stage. This pattern suggests an EPHA7 role in fetal pro/pre-B cell expansion and/or differentiation [93]. We should note that there are different EPHA7 splice variants, encoding either the full-length EPHA7 or a truncated soluble form lacking the cytoplasmic domain. Taking this into consideration, Dawson et al. demonstrated that mature B cells express and secrete the truncated form of EPHA7, which blocks ephrinA4-mediated activation of the full-length EPHA7 and partially inhibits chemokine-induced B cell migration [94]. Activated B cells in tonsils can secrete a splice variant of the human ephrinA4 encoding a soluble form of the protein, with a possible implication in the interaction between activated B cells and dendritic cells (DCs) [113]. B cell activation leads to EPHB2 upregulation, and its expression level positively correlates with B cell proliferation, tumor necrosis factor (TNF)-α secretion, and immunoglobulin (Ig)G production. Its effects have been shown to be mediated through the Src-p65 and Notch1 signaling pathways and could be regulated by miR-185 [114]. Laidlaw et al. identified ephrinB1 as a marker of germinal center (GC) and memory precursor B cells. During an immune response, there is an increase in the number of ephrinB1+ GC B cells, which express elevated levels of *Bcl6*, *S1pr2*, and *Aicda* relative to their ephrinB1- counterparts. A small subpopulation of recently dividing, somatically mutated GC B cells that have begun to down-regulate *Bcl6* and *S1pr2* and express memory B cell markers, such as CD38 and Epstein-Barr virus-induced gene 2 (EBI2), likely representing a population of GC memory precursor B cells, was also positive for ephrinB1 [115]. The expression of ephrinB1 by GC B cells is implicated in the control of T follicular helper (TFH) cell trafficking and function. Interactions between ephrinB1 expressed by B cells and EPHB6 expressed by T cells repulsively inhibit T cell-to-B cell adhesion and lead to decreased T cell retention in the GC. At the same time, ephrinB1 signaling, predominantly through EPHB4, also expressed in T cells, promotes IL-21 production from GC TFH. These interactions are crucial for normal plasma cell formation since it has been shown that in the absence of ephrinB1, TFH cells abnormally accumulate in the GC and show reduced IL-21 production, leading to decreased plasma cell production [116].

Regarding T cell lymphopoiesis, the expression of multiple EPHs (i.e., EPHA1, EPHA2, EPHA3, EPHA4, EPHA7, EPHB2, and EPHB6) and ephrins (i.e., ephrinA1, ephrinA3, ephrinA5, and ephrinB1) has been described in the thymus on both developing T cells and thymic stromal cells [88,98,99]. Moreover, multiple processes of T cell thymic education and functional maturation are under the control of EPH/ephrin-mediated interactions between lymphoid cells and thymic epithelial cells [117,118]. EPH/ephrin signaling also plays a role in the context of T cell transendothelial migration. Aasheim et al. showed that EPHA1 and EPHA4 expressed on CD4+ T cells interact with ephrinA1 expressed on high endothelial venule (HEV) endothelial cells, promoting chemotaxis, intracellular proline-rich tyrosine kinase 2 (PYK2) phosphorylation, and actin polymerization [119]. This effect seems to be mediated by stromal cell-derived factor 1a (SDF-1a) [119,120]. In mice, PB T cells express ephrinB3 and its receptors, which have been shown to be implicated in T cell/T cell and T cell/antigen-presenting cell interactions and to promote T cell activation and function [121]. EphrinB1 was also shown to be pivotal for T cell/T cell co-stimulation and the reduction of the T cell response threshold to antigen stimulation [122]. Finally, EPH/ephrin signaling regulates integrin-mediated adhesive T cell properties in different modes since integrin-mediated adhesion is stimulated by ephrinA activation, whereas it is inhibited by EPHA signaling [123].

### 3.4. Other Hematopoietic and Immune Cells

With regards to platelets, EPH/ephrin signaling affects their functionality, namely the processes of adhesion, activation, and aggregation, which in turn define thrombus stability, with the implicated molecules being EPHA4, EPHB1, EPHB2, and ephrinB1 [91,92,124,125]. Moreover, a c.2233C > T variant (missense p.R745C) of the *EPHB2* gene has been associated with defects in platelet aggregation, αIIbβ3 activation, and granule secretion induced by G-protein-coupled receptor (GPCR) agonists and convulxin, as well as in thrombus formation on collagen under flow [126].

Multiple EPHs (i.e., EPHA2, EPHA4, EPHB2, EPHB4) and ephrins (i.e., ephrinA1, ephrinA2, ephrinA4, and ephrinB2) seem to be involved in the processes of adhesion, migration, and extravasation of monocytes and macrophages (reviewed in [12]), and ephrinA4 expression has been shown to be upregulated during monocyte differentiation [127].

Expression of EPHA2, EPHB1, EPHB2, and EPHB3 by DCs has also been reported and has been mainly correlated with the functions of cell organization and trafficking [128,129,130].

Figure 2 summarizes the expression of ephrins and EPHs as well as their reported interactions during normal hematopoiesis.

## 4. EPH/Ephrin Expression in the Setting of Hematologic Neoplasia

The intricate role of EPH/ephrin signaling in the pathogenesis of hematologic neoplasia has been studied in vitro with the use of leukemia/lymphoma cell lines as well as in patients with various hematologic malignancies. Table 2 summarizes the reported expression and aberrations at genetic and epigenetic levels of ephrins and EPHs in leukemia/lymphoma cell lines and hematologic neoplasms.

### 4.1. EPH/Ephrin Expression in Leukemia/Lymphoma Cell Lines

The expression of EPHs and ephrins has been studied in various leukemia/lymphoma cell lines, unraveling their possible implication in the pathogenesis of hematologic malignancies. In 1999, Steube et al. investigated the expression, at the transcriptional level, of EPHB4 and its ligand ephrinB2 in 70 leukemia/lymphoma cell lines. They identified EPHB4 mRNA expression in 68/70 cell lines and ephrinB2 expression in 58/70 cell lines, highlighting a possible role of EPH/ephrin signaling in hematologic neoplasms [143].

Regarding malignant cell lines of myeloid origin, ephrinB2 was shown to induce proliferation and colony formation in an erythroid leukemia-derived cell line, also upregulating the expression of growth-related genes such as *FBJ* murine osteosarcoma viral oncogene homolog B and *v-src* avian sarcoma viral oncogene homolog [156]. Using a human imatinib-resistant chronic myelogenous leukemia (CML) cell line that overexpressed EPHB4 (K562 R), Li et al. showed that EPHB4 knockdown inhibited the migration ability of the cells and restored their sensitivity to imatinib in vitro. This restored imatinib sensitivity was concurrent with increased phosphorylation levels of EPHB4 and decreased phosphorylation levels of RhoA, Rac1, and Cdc42. Based on these findings, the authors concluded that the aberrant activation of EPHB4/ephrinB2 may mediate CML resistance to imatinib, possibly via the involvement of cytoskeletal proteins [144].

More abundant are the studies of EPH/ephrin signaling in leukemia/lymphoma cell lines of lymphoid origin. EPHA3 was originally discovered as a receptor in the LK63 pre-B acute lymphoblastic leukemia (ALL) cell line [132], while its expression has also been identified in other leukemic cell lines, namely the T cell leukemia cell lines Jurkat, JM, and HSB-2 [133]. Wimmer-Kleikamp et al. showed that after exposure to ephrinA5 surfaces, EPHA3-positive LK63 pre-B ALL cells tended to convert from a suspended/nonpolarized cell type to an adherent/polarized cell type. This transition relied on EPHA3 functions operating in the absence of EPH signaling. The morphological changes and adhesion of LK63 cells were attenuated by the endogenous protein tyrosine phosphatase (PTP) activity, while PTP inhibition and effective EPHA3 signaling reversed the phenotype of non-adherent cells with a condensed cytoskeleton. Altogether, these findings suggest that ephrin contact can be modified from repulsion to adhesion by EPH-associated PTP activities, highlighting a possible role in the pathology of B cell hematologic malignancies [157].

The study of EPH/ephrin signaling in leukemic T cell lines led to the identification of EPH/ephrinA interactions as a possible player in the promotion of malignant cell growth. *EPHA3* was identified as a CD28-responsive gene in Jurkat cells. Stimulation of Jurkat cells by ephrinA5 diminished their adhesion to fibronectin and led to the recruitment of the adapter protein CrkII to EPHA3. EPHA3 expression in CD28-stimulated Jurkat cells was enhanced by insulin-like growth factor 1 (IGF-1) and overexpression of the IGF-1 receptor (IGF-1R), while anti-IGR-1R blocking antibodies suppressed EPHA3 expression. These data suggested that regulation of *EPHA3* expression by CD28 and IGF-1 correlates with cell adhesion and may be involved in the motility of malignant T cells [134]. Moreover, phosphorylation of several kinases, such as Lck, Fyn, and Akt, was proven to be stimulated by ephrinA signaling, with subsequent inhibition of antigen receptor-induced apoptosis [158]. Overall, these results indicate the role of ephrinA signaling both in the motility and survival of malignant T cells.

EphrinB signaling has also been shown to contribute to the survival and regulate the invasiveness of neoplastic T cells. Studies in the T-ALL cell lines H9 and E6.1 demonstrated that EPHB3 is consistently expressed, most frequently in combination with EPHB6. Stimulation of these receptors by EphrinB1 and EphrinB2 potently suppressed Fas-induced apoptosis. This effect has been associated with Akt activation and inhibition of the Fas receptor-induced caspase proteolytic cascade [73]. Jian et al. highlighted that ephrinB1 can regulate T leukemia cells’ functions through the control of lipid-raft-associated signaling, adhesion, and invasive activity. Specifically, after treatment of Jurkat cells with EPHB3, they reported enhanced ephrinB1 phosphorylation and its relocalization into lipid drafts, an event mediated by the T lineage-specific kinase, Lck. EphrinB1 was also shown to drive the colocalization of the cytoskeleton regulators CrkL and Rac1 and to induce a strong repulsive response in leukemic cells. Moreover, the metastatic properties of leukemic cells can be amplified by ephrinB1, given that its suppression inhibits the leukemic cells’ invasiveness [154]. Suzuki et al. suggested a new mechanism for imatinib resistance in Philadelphia chromosome-positive ALL cell lines mediated by the activation of the RAS/mitogen-activated protein kinase (MAPK) pathway and EPHB4 [145].

The role of EPH/ephrin signaling has also been explored in leukemic cell lines with *ALL1* gene translocations. The *ALL1* gene (otherwise known as *MLL*) is involved in recurrent chromosome translocations identified in 5–10% of acute leukemias, both of myeloid and lymphoid origin [159]. The result of these translocations is the fusion of the *ALL1* gene with >80 partner genes and the production of leukemogenic proteins constituting the N-terminal ALL1 sequence and the C-terminal sequence of the partner protein [160]. The most common translocations in *ALL1*-associated leukemias are t(4;11) and t(9;11), which generate the ALL1/AF4 and ALL1/AF9 fusion proteins, respectively [161]. Prompted by the observation that only fetal BM pro-B and pre-B cells, and not adult B-lineage cells, express EPHA7 [93], Nakanishi et al. studied the expression of EPHA7 and its ligands in *ALL1*-associated leukemias, trying to determine whether ALL1 fusion proteins are involved in the regulation of the EPH and ephrin genes. The authors demonstrated that K562 cells producing recombinant ALL1/AF4 or ALL1/AF9 fusion proteins were characterized by transcriptional up-regulation of *EPHA7*. Consistently, siRNA-mediated suppression of *ALL1/AF4* in SEMK2 cells carrying the t(4;11) chromosome translocation resulted in the down-regulation of EPHA7. The demonstrated occupancy of tagged ALL1 fusion proteins on EPHA7 pointed to EPHA7 as a direct target of ALL1 fusion proteins. Furthermore, EPHA7 upregulation was concurrent with ERK phosphorylation, whereby inhibition of ERK phosphorylation led to apoptotic cell death specifically in leukemic cells carrying the t(4;11) chromosome translocation [139].

### 4.2. EPH/Ephrin Expression in Hematologic Malignancies

#### 4.2.1. Leukemias of Myeloid Origin

EPHAs’ aberrations at the genetic, transcriptional, and protein levels have been documented in AML patients. Copy number alterations of the *EPHA3* gene have been identified in the genomes of patients diagnosed with AML [137]. In a cohort of 58 AML patients (39 adults and 19 children), *EPHA4* expression, assessed by RT-PCR in PB samples, was positive in 56.4% of adult and 36.8% of pediatric AML patients [162]. Lower serum levels of EPHA4 have been reported in de novo AML patients compared to healthy controls (*p* < 0.0001), while in the same study, a significant positive correlation between EPHA4 and overall survival (OS) (*p* = 0.007) was also demonstrated [163].

Regarding type B ephrins and EPHs, Merchant et al. reported that 28% (7/25) of BM samples from newly diagnosed AML patients examined in their study overexpressed EPHB4. This expression was specific for leukemic cells since EPHB4 expression was undetectable in non-leukemic CD34+ HSPCs. The authors demonstrated that the effects of EPHB4 signaling are mediated by the PI3K/AKT pathway since knockdown of *EPHB4* inhibits PI3K/AKT signaling, leading to reduced cell viability. This phenomenon could be rescued by a constitutively active form of AKT [146]. Increased *EPHB6* expression has also been reported in AML [149]. On the other hand, in another study, BM mononuclear cells from pediatric AML patients showed suppressed levels of EPHB1 phosphorylation and mRNA expression compared to healthy controls, while a significant percentage of the AML specimens presented hypermethylation of the *EPHB1* promoter. In the presence of EPHB1, ephrinB1-mediated signaling led to p53 DNA binding, with a subsequent restoration of the DNA damage response (DDR) cascade by the activation of ataxia-telangiectasia mutated (ATM)- and Rad3-related (ATR), Chk1, p53, p21, p38, cyclin dependent kinase 1 (CDK1) (tyr15), and Bax, and downregulation of heat shock protein (HSP)27 and Bcl2. The reintroduction of EPHB1 expression in EPHB1-methylated AML cells enhanced the same cascade of ATR, Chk1, p21, and CDK1(tyr15), which in turn enhanced programmed cell death. Therefore, in pediatric AML, a tumor-suppressor function of EPHB1 could be proposed via its effect on the DDR system [142].

Li et al. demonstrated that in BM cells from CML patients, EPHB4 mRNA levels significantly increased according to clinical stages (higher EPHB4 expression for CML in blast crisis compared to CML in chronic phase). The level of EPHB4 mRNA expression also correlated with complete cytogenetic response (CCyR) after 12 months of imatinib treatment, with low expression being associated with higher rates of CCyR [72.7% CCyR for the low EPHB4 group vs. 36.3% CCyR for the high EPHB4 group (*p* < 0.01)] [144]. The results of these studies propose a possible use of EPHB4 mRNA levels both as a biomarker of CML clinical stage and as a predictive factor of response to imatinib treatment.

Cooperation response genes (CRGs) are targets for synergistic regulation by oncogenes and have been demonstrated to define cancer phenotypes [164]. Using a mouse model of the CML blast crisis, Asthon et al. identified *EPHA3* as a common CRG. In this model, BM HSCs were infected with retroviral vectors encoding the human fusion genes *BCR-ABL* and *NUP98-HOXA9*, found in human myeloid leukemias. *EPHA3* knockdown by shRNA in leukemic stem cells (LSC) led to reduced engraftment, highlighting the implication of this gene in LSC growth and survival in the BM microenvironment [135]. In a study including BM samples from patients diagnosed with a variety of hematologic malignancies, both of myeloid and lymphoid origin, copy number variations (CNVs) of *EPHA3* were associated with ALL, AML, CLL, CML, MM, and myelodysplastic neoplasms (MDS). The mRNA levels of EPHA3 correlated weakly yet positively with the copy numbers of EPHA3 [136].

#### 4.2.2. Leukemias of Lymphoid Origin

In an effort to define differentially expressed genes (DEGs) in pediatric B precursor ALL, *EFNB1* was identified among the downregulated DEGs enriched in the cell cycle process [165]. More recently, Xiang et al. constructed a 5-year OS prognostic risk model for children and young adults with B-ALL. The authors compared DEGs between normal B cells and B-ALL cells and proposed a three-gene signature including *EFNB1*, along with *CYBB* and *BCL2A1*, as an independent prognostic factor for B-ALL. *EFNB1* mRNA levels were significantly lower in the B-ALL group compared to controls [155]. While normally EPHB6 is expressed by a very small population of human PB CD4+ or CD8+ T cells and about 1.5% of normal BM lymphocytes, human leukemia/lymphoma cells of T origin demonstrate conserved or increased EPHB6 expression [150,151]. Of note, the expression level of EPHB6 decreases with the maturation of the cells in T cell-derived leukemia cells, therefore suggesting that EPHB6 expression regulates T cell development but has a less significant role in mature T cells [150].

Of interest, certain EPHs were proposed to have a tumor suppressive role in the setting of ALL. Aberrant methylation and epigenetic inactivation of EPH and ephrin genes have been identified in ALL. Li et al. reported a prevalent methylation of the *EPHB4* gene in childhood ALL (the patient cohort included both B-ALL and T-ALL cases, with B-ALL predominance), possibly resulting in expressional inactivation and consequent promotion of ALL progression. An association with an unfavorable prognosis has also been observed. Therefore, the authors proposed that the *EPHB4* gene may function as a potential tumor suppressor in childhood ALL [147]. Another study by Kuang et al. identified hypermethylation of multiple EPH and ephrin genes (i.e., *EPHA2*, -*A4*, -*A5*, -*A6*, -*A7*, -*A10*, *EPHB1*, -*B2*, -*B3*, -*B4*, *EFNA1*, -*A3*, -*A5*, and *EFNB1* and -*B2*) in leukemia cell lines and primary ALL BM samples. Expression analysis of *EPHB4*, *EFNB2*, and *EFNA5* genes demonstrated that DNA methylation was associated with gene silencing. This study also proposed *EPHB4* as a possible tumor suppressor gene since induction of *EPHB4* expression by lentiviral transduction in Raji cells, in which *EPHB4* is methylated and silenced, resulted in reduced proliferation and apoptotic cell death [131]. EPHA7 has been proposed as a tumor suppressor implicated in the pathogenesis of T-ALL/lymphoma with 6q deletion [140].

The expression of EPHs has also been correlated with sensitivity or resistance to drugs used for ALL. In T-ALL, EPHB6 expression, via its effect on the Akt pathway, has been shown to confer increased sensitivity to doxorubicin. Specifically, EPHB6 suppressed Akt signaling in doxorubicin-treated T-ALL cells via a decrease in the phosphorylation of Akt and its downstream target, the p70 S6 kinase [151].

EPH/ephrin expression has been further studied in chronic lymphocytic leukemia (CLL) cases. EPHs and ephrins, such as ephrin A4, EPHB6, and EPHA10, are frequently expressed by PB CLL B cells, whereas in normal individuals their expression is confined to lymph nodes (LNs) and activated B-cells. EPHB6 expression was correlated with a high level of ZAP-70 mRNA and a poor prognosis. High serum levels of a soluble ephrinA4 isoform were positively correlated with increasing PB lymphocyte counts and lymphadenopathy [152], while the levels of ephrinA4 expression on PB CLL cells were inversely correlated with lymphadenopathy [153].

In CLL, it has been proposed that EPHA2-ephrinA4 interaction in the LN HEVs could mediate leukocyte trafficking between blood and tissues [166]. The observed reduced extravasation capacity of CLL cells compared to normal B cells could be partially explained by ephrinA4 overexpression by the former, which could possibly induce repulsive signals. CLL cells from patients with lower levels of ephrinA4 are associated with better in vitro adhesion and transendothelial migration (TEM) compared to CLL cells from patients without lyphadenopathy. Both adhesion and TEM are critical steps of the extravasation process and can be modulated via the interaction of ephrinA4 expressed on lymphocytes with EPHA2 expressed on the luminal side of CD31+ endothelial cells of the LN vasculature [153].

Flores et al. further proposed that CLL cells from patients with lymphadenopathy demonstrate a survival advantage during TEM, possibly associated with increased expression of α4 and αL integrin chains. The authors proposed that integrins are sequestered and inactivated by ephrinA4 expressed on the surface of CLL cells, resulting in reduced adhesion and inhibition of survival signals. We should note that *EFNA4* silencing induced increased survival only for CLL cells from patients with lymphadenopathy but not from those without this clinical feature. The addition of a soluble ephrinA4 isoform led to similar results, suggesting that the accumulation of this isoform in the serum of CLL patients with lymphadenopathy contributes to the dissemination and survival of CLL cells in vivo. In cases of CLL lymphadenopathy, apoptotic CLL cells lacking ephrinA4 preferentially accumulate around HEVs. Additionally, soluble ephrinA4 isolated from the sera of patients led to an increase in the number and viability of CLL cells recovered from the LNs of adoptively transferred mice. Overall, these results indicate that ephrinA4-mediated survival during TEM could enhance the transcellular TEM route of the CLL cells [167].

#### 4.2.3. Lymphomas

As opposed to the observed EPHA7 upregulation in acute leukemias [139], *EPHA7* hypermethylation and repression have been identified in mouse and human GC B cell non-Hodgkin lymphomas (NHLs), with the potential to influence tumor progression and spread [94]. EPHA7 has been proposed as a possible tumor suppressor of follicular lymphoma (FL) since deletions in 6q11-27 commonly found in FL lead to the loss of the *EPHA7* gene [141].

Huang et al. demonstrated that EPHB4 expression was suppressed in Epstein-Barr virus (EBV)-positive post-transplant lymphoproliferative disorders (PTLDs). After EBV infection, transcriptional and translational EPHA4 downregulation, mediated by the EBV-encoded latent membrane protein 1 (LMP1), is observed in primary B cells. Functionally, it seems that EPHA4 expression prevents the proliferation of transformed B cells after EBV infection. Apart from PTLDs, an inverse correlation between EPHA4 expression and EBV positivity was also identified by immunochemical staining in the setting of EBV+ and EBV- diffuse large B cell lymphoma (DLBCL), with lower EPHA4 expression being associated with a poor survival rate in DLBCL patients [148].

#### 4.2.4. Multiple Myeloma (MM)

EPHA3 is highly expressed in endothelial BM cells from patients with MM and has been implicated in MM angiogenesis. It has been shown that siRNA targeting of EPHA3 inhibited MM endothelial cells from adhering to fibronectin, migrating, and forming tube-like structures in vitro. Moreover, *EPHA3* knockdown resulted in the downregulation of molecules implicated in adhesion, migration, and invasion processes. Interestingly, an anti-EPHA3 antibody reduced all the MM endothelial cell angiogenesis-related functions in vitro [138].

Figure 3 illustrates EPH/ephrin aberrations that have been studied in different types of hematologic malignancies, as well as proposed signaling pathways and functions implicated in disease pathogenesis.

## 5. EPH/Ephrin Therapeutic Targeting in Hematologic Malignancies

EPHA2 over-expression has been documented in a mouse model of AML expressing the ALL1/AF9 fusion protein [168] and, as analyzed before, in human leukemias with *ALL1* translocations [139]. Though Charmsaz et al. demonstrated that EPHA2 does not contribute to the initiation and progression of leukemia using *EPHA2* knockout mice with ALL1/AF9 leukemias, they investigated its role as a possible therapeutic target. As expected, administration of the EPHA2 monoclonal antibody (mAb) IF7 alone did not affect leukemogenesis. However, when IF7 was used to deliver a radioactive “payload” (Lutetium 177) to the leukemic cells, a survival advantage was noted for the animals treated with the drug compared to controls [168].

In models of EPHA3-expressing leukemic xenografts, EPHA3 monoclonal antibody therapy resulted in a significant response, while the therapeutic effect was greatly enhanced by adding an α-particle-emitting Bismuth-213 payload [169]. Taking into consideration that EPHA3 is overexpressed in many hematologic malignancies, a phase I study investigated the safety and tolerability of KB004, an anti-EPHA3 mAb. The study included patients with AML (*n* = 39), myelodysplastic/myeloproliferative neoplasms (MDS/MPN) (*n* = 3), MDS (*n* = 4), DLBCL (*n* = 1), and myelofibrosis (MF) (*n* = 3). Transient grade 1 and 2 infusion reactions were described as the most common toxicities, while among the different disease settings, the authors stated that responses were observed in patients with AML, MF, MDS/MPN, and MDS [170].

Based on the study of Nakanishi et al., targeting EPHA7 may have a place in the treatment of acute leukemias with translocations involving *ALL1*. The authors proposed that ALL1 fusion proteins directly upregulate EPHA7 expression, which in turn leads to ERK phosphorylation. Treatment of the leukemic cells with an inhibitor of ERK phosphorylation induced apoptotic cell death [139]. Overall, these findings indicate a therapeutic role for EPHA7 inhibition, possibly in combination with ERK signaling inhibitors, in leukemias with *ALL1* translocations. On the other hand, restoration of EPHA7 tumor-suppressor properties could be exploited within the spectrum of therapeutic approaches for lymphomas. The absence of EPHA7 expression has been documented in lymphomas due to *EPHA7* promoter hypermethylation [94] or *EPHA7* deletion [141]. Soluble EPHA7 binds to EPHA2 and blocks its activity and downstream signaling, including the pathways of ERK and SRC kinases. Notably, treatment of cultured or xenografted lymphoma cells with EPHA7 resulted in ERK and SRC inhibition and cell death, a finding suggestive of a potential therapeutic application for EPHA7 in lymphoma therapy [141]. Therefore, to achieve the delivery of therapeutically effective EPHA7 concentrations to lymphomas in vivo, Orrichio et al. used the anti-CD20 mAb Rituximab. The efficacy of this approach relied on the combination of the intrinsic anti-tumor effect of the anti-CD20 mAb with the specific tumor suppressor effect of EPHA7, exploiting its ability to shut down ERK and SRC signaling in lymphoma cells [141].

TPD7, a novel biphenyl urea taspine derivative, was shown to inhibit cancer growth by targeting vascular endothelial growth factor receptor 2 (VEGFR2) [171]. EphrinB2, via reverse signaling mediated by PDZ domain interactions, regulates the internalization and signaling activity of VEGFR2 [172]. Berberine, a nitrogenous cyclic plant-derived compound, led to cancer cell growth inhibition by targeting ephrinB2 and VEGFR2 signaling pathways [173,174]. Ma et al. demonstrated that treatment of Jurkat cells with the combination of TPD7 and berberine (TAB) resulted in a significant decrease in ephrinB2 and its phosphorylation levels, while it also downregulated PDZ domain-interacting proteins [syntenin-1 and protein interacting with C Kinase-1 (PICK1)], an effect that correlated with decreased phosphorylation of VEGFR2. Apart from regulating ephrinB2 signaling, the TAB combination was also shown to modulate related signal transduction pathways, as proven by the inhibition of Rac1 and upregulation of the phosphatase and tensin homolog (PTEN). Therefore, the TAB combination may be a promising therapeutic approach for the treatment of T-cell ALL [175].

The targeting of EPHB4 has gained attention in cases of drug-resistant leukemias. The sensitivity of myeloid leukemia cells to Adriamycin in vitro has been shown to be associated with EPHB4 expression. The stimulation of EPHB4 by EphrinB2-Fc sensitized HL60/ADM cells to Adriamycin in a dose-dependent manner, leading to reduction of tumor cell viability and induction of apoptosis. This observation provides proof for the development of a therapeutic approach targeting EPHB4 in the setting of drug-resistant leukemias [176]. In the case of CML, inhibition of EPHB4 to overcome imatinib resistance has been explored. In K562 cells, vandetanib demonstrated a stable interaction with a hydrophobic pocket of the intracellular kinase domain of EPHB4, while it also downregulated EPHB4 via the ubiquitin-proteasome pathway and inhibited the PI3K/AKT and MAPK/ERK signaling pathways. The combination of vandetanib and imatinib exhibited enhanced and synergistic growth inhibition against imatinib-resistant K562 cells in vitro and in vivo. These findings suggest that vandetanib drives growth arrest and overcomes the resistance to imatinib in CML via EPHB4 targeting [177]. Apart from the setting of drug resistance, a highly specific mAb against EPHB4 (MAb131) was shown to be effective against AML in vitro and in vivo, rendering EPHB4 a potential therapeutic target for the subset of AML cases with high EPHB4 expression [146].

Table 3 sums up the therapeutic targeting approaches of the EPH/ephrin axis in the setting of hematologic malignancies.

## 6. Discussion

The expression of EPHs and ephrins by normal hematopoietic cells is rather heterogenous and varies based on the stage of differentiation, the lineage commitment, and the hematopoietic compartment. EPH/ephrin signaling mainly defines the interactions of hematopoietic cells with other cells in their microenvironment as well as those between different subpopulations of hematopoietic cells, contributing to their maintenance, maturation, and function. We should note that there are still many aspects of this complex signaling pathway to be deciphered in the context of normal hematopoiesis and even more in the context of hematologic malignancies.

Aberrations in EPH/ephrin expression and signaling might lead to perturbations of the normal HSC pool, thus contributing to the pathogenesis of hematologic neoplasia. The interaction between EPHA3 expressed by HSCs and ephrinA5 was shown to contribute to HSC homing in BM transplantation mouse models [90], while EPHA3 expression in leukemic cells was associated with increased adherence properties of the leukemic cells [157]. In this setting, it can be proposed that the detected CNVs leading to *EPHA3* loss in AML [137] and the identification of *EPHA3* as a CRG in the CML blast crisis might affect BM localization and trafficking of blast cells. EPHB1 expressed on HSPCs induces ephrinB1 reverse signaling in BM stromal cells, a process necessary for CXCL12 production and thus subsequent regulation of HSPC function [107]. Loss of EPHB1 expression can be responsible for abnormal HSPC function, possibly explaining an aspect of the pathophysiology of pediatric AML where hypermethylation of the *EPHB1* promoter leads to decreased EPHB1 expression [142].

The diverse expression of EPHs and ephrins in different hematologic malignancies can be interpreted under the prism of the contribution of the EPH/ephrin signaling axis in HSC differentiation to lineage-committed progenitors. EPHB4, a receptor identified in HSCs and associated with myeloid differentiation [100,101], presents different patterns of expression in myeloid and lymphoid malignancies. Namely, it has been demonstrated to be over-expressed in AML and CML blast crises [144,146], while *EPHB4* is hypermethylated in ALL [131,147], and EPHB4 expression is suppressed in EBV+ PTLDs [148].

Of interest, even though research has identified certain types of EPHs and ephrins to be implicated in the normal development and function of certain hematopoietic lineages, the same EPHs and ephrins have been associated with hematologic malignancies derived from different hematopoietic cell subtypes. A characteristic example is that though the role of EPHB6 during normal hematopoiesis has mainly been studied in T cells, specifically in thymic T cell development [98] and T cell-B cell interactions [116], its increased expression has not only been identified in T-cell leukemia/lymphoma [150,151] but also in AML cases [149]. Another paradigm is that of ephrinB1, which has been so far only studied in mouse normal thymic T cell development [99], while research in human B-ALL also suggests its role in human B lymphopoiesis since *EFNB1* is downregulated in B-ALL cases, where it can also serve as an independent prognostic factor [155].

Furthermore, in different disease settings, EPHs could either favor or inhibit neoplastic progression. In the case of hematopoietic malignancies, EPHA7 is a typical example. On the one hand, leukemic cell lines with *ALL1* translocations are characterized by transcriptional up-regulation of *EPHA7* [139], while on the other hand, EPHA7 has been proposed as a possible tumor suppressor in FLs [141].

In addition, different clinical phenotypes of malignancy could be correlated with different expression patterns and/or isoforms of ephrins. CLL serves as such a paradigm, where high serum levels of a soluble ephrinA4 isoform are positively correlated with increased PB lymphocyte counts and lymphadenopathy [152], while ephrinA4 expression on PB CLL cells is inversely correlated with lymphadenopathy [153]. Given that the soluble form of ephrinA4, normally secreted by activated B cells, seems to be implicated in B cell-DC interactions [113], its aberrant expression in the case of CLL might contribute to B cell retention in the LNs and affect B cell trafficking to the PB.

We should also note that the study of EPH/ephrin expression in hematologic malignancies could convey a possible role in disease prognosis. Thus far, decreased expression of *EFNB1* has been identified as an independent prognostic factor for B-ALL [155], and EPHB6 expression was shown to be an adverse prognostic marker in CLL [152].

Finally, in the therapeutic setting, the expression of specific types of EPHs, or ephrins, could be exploited for targeting specific types of hematologic malignancies. As an example, EPHA7 could be an attractive target for leukemias with *ALL1* translocations [139]. We should highlight that the study of the EPH/ephrin signaling axis should not only be limited in its therapeutic targeting, which is still mainly at preclinical stages. The levels of EPH/ephrin expression could also serve as a predictive indicator of response to commonly administered drugs. Namely, EPHB6 expression levels could be used as a predictive marker of sensitivity to doxorubicin in T-ALL [151], while EPHB4 overexpression in CML and increased activation in Philadelphia chromosome-positive ALL could be used as surrogate biomarkers of imatinib resistance [144,145]. Further research to identify key players of the EPH/ephrin axis in each type of hematologic neoplasm is indispensable to achieving proper therapeutic targeting and accelerating the development of new drugs for application in clinical trials.

## 7. Conclusions

To conclude, given the diversity and complexity of EPH/ephrin signaling in the process of normal hematopoiesis as well as in the pathophysiology of various hematologic malignancies, further research is necessary to unravel its implications in the pathogenesis of the different subtypes of hematologic neoplasia. Deciphering the network of EPH/ephrin signaling pathways could contribute to the identification of possible new therapeutic targets for these challenging-to-treat types of malignancies.

## Figures and Tables

**Figure 1 cancers-15-03963-f001:**
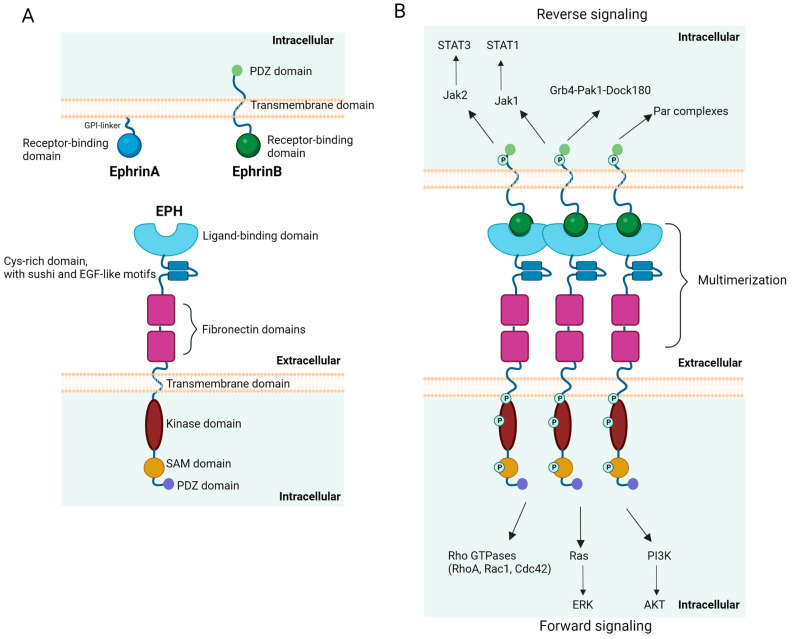
Schematic representation of the structure (**A**) and signaling (**B**) of ephrins and EPHs. The main signaling pathways induced by the EPH/ephrin interaction are shown. EGF, epidermal growth factor; ERK, extracellular signal-regulated kinase; GPI, glycosylphosphatidylinositol; Jak, Janus kinase; PDZ, post-synaptic density protein (PSD95), Drosophila disc large tumor suppressor (Dlg1), and zonula occludens-1 protein; P, phosphorylated; PI3K, phosphoinositide 3 kinase; SAM, sterile alpha motif; STAT, signal transducer and activator of transcription. Created with BioRender.com (accessed on 1 August 2023).

**Figure 2 cancers-15-03963-f002:**
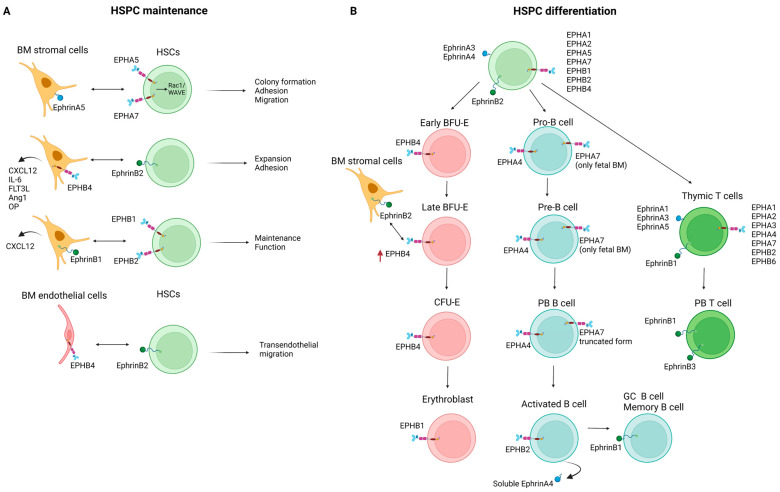
Expression and interactions of ephrins and EPHs during normal hematopoiesis, focusing on their implication in HSPC maintenance (**A**) and differentiation (**B**). Ang1, angiopoietin 1; BFU-E, burst-forming units-erythroid; BM, bone marrow; CXCL12, C-X-C motif ligand 12; FLT3L, Fms-like tyrosine kinase 3 (FLT3) ligand; GC, germinal center; HSCs, hematopoietic stem cells; HSPC, hematopoietic stem/progenitor cell; IL-6, interleukin 6; OP, osteopontin; PB, peripheral blood. Created with BioRender.com (accessed on 1 August 2023).

**Figure 3 cancers-15-03963-f003:**
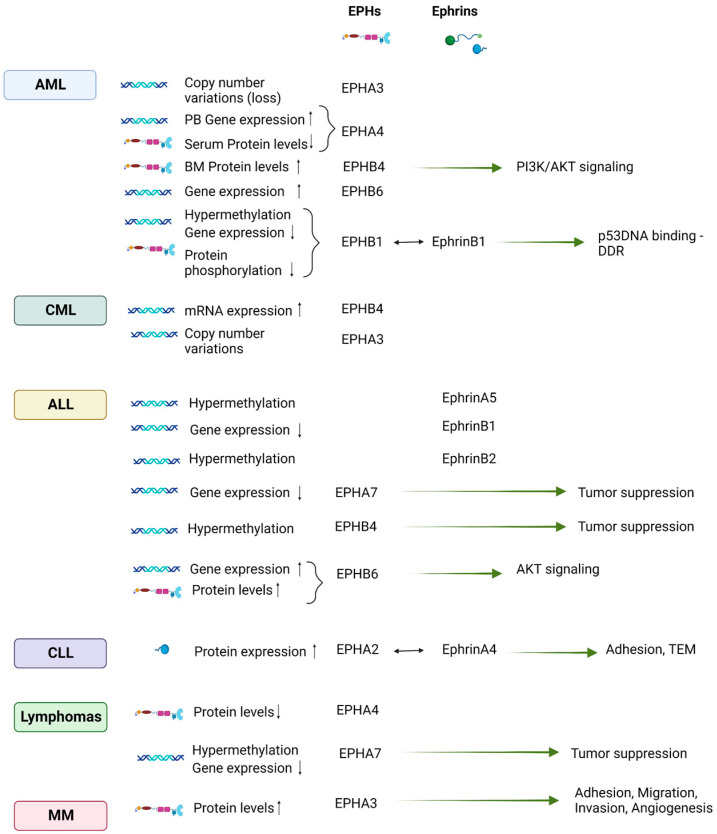
Ephrin/EPH aberrations in different types of hematologic malignancies and proposed signaling pathways and functions implicated in disease pathogenesis. BM, bone marrow; DDR, DNA damage response; PB, peripheral blood; PI3K, phosphoinositide 3 kinase; TEM, transendothelial migration. Bidirectional arrows signify interaction. Unidirectional green arrows signify induction of a function. Created with BioRender.com (accessed on 1 August 2023).

**Table 1 cancers-15-03963-t001:** Expression of ephrins and EPHs in human and murine normal hematopoietic cells.

EPHA1	Human CD34+ HSCs [86,87] Rat T lymphocytes—during thymic development [88]
EPHA2	Human CD34+ HSCs [89] Human CD133+ HSCs [89] Mouse Lin-ckit + sca1+ HSCs [90] Rat T lymphocytes—during thymic development [88]
EPHA3	Mouse Lin-ckit + sca1+ HSCs [90] Rat T lymphocytes—during thymic development [88]
EPHA4	Human platelets [91,92] Rat T lymphocytes—during thymic development [88] Human pro- and pre-B cells [93]
EPHA6	Mouse Lin-ckit + sca1+ HSCs [90]
EPHA7	Rat T lymphocytes—during thymic development [88] Human pro- and pre-B cells [93] Normal lymphocytes [94]
EPHA8	Mouse Lin-ckit + sca1+ HSCs [90]
EPHB1	Human platelets [92]
EPHB2	Human CD133+ HSCs [89] Human CD34+ HSCs—partially expressed [89]
EPHB4	Bone marrow CD34+ cells [95] Erythroid progenitor cells [96] HSPCs [97]
EPHB6	Mouse thymic T cells [98]
EphrinA1	Rat T lymphocytes—during thymic development [88]
EphrinA3	Human CD34+ HSPCs [87] Rat T lymphocytes—during thymic development [88]
EphrinA4	Human CD34+ HSPCs [87] Mouse Lin-ckit + sca1+ HSCs [90]
EphrinA5	Mouse Lin-ckit + sca1+ HSCs [90] Rat T lymphocytes—during thymic development [88]
EphrinB1	Human platelets [92] Mouse T lymphocytes—during thymic development [99]

HSCs, hematopoietic stem cells; HSPCs, hematopoietic stem/progenitor cells.

**Table 2 cancers-15-03963-t002:** EPH/ephrin genetic and epigenetic aberrations and expression in leukemia/lymphoma cell lines and hematologic malignancies.

	Leukemia/Lymphoma Cell Lines	Hematologic Malignancies
EPHA2		ALL: *EPHA2* hypermethylation [131]
EPHA3	LK63 pre-B ALL cell line [132] T-cell leukemia cell lines (Jurkat, JM, HSB-2) [133] *EPHA3* identified as a CD28-responsive gene in Jurkat cells [134]	Mouse model of CML blast crisis: *EPHA3* identified as a common CRG [135] AML: *EPHA3* CNVs [136,137] ALL: *EPHA3* CNVs [136] CLL: *EPHA3* CNVs [136] CML: *EPHA3* CNVs [136] MDS: *EPHA3* CNVs [136] MM: EPHA3 highly expressed in endothelial BM cells [138]
EPHA4		ALL: *EPHA4* hypermethylation [131]
EPHA5		ALL: *EPHA5* hypermethylation [131]
EPHA6		ALL: *EPHA6* hypermethylation [131]
EPHA7	Leukemic cell lines with *ALL1* gene translocations: transcriptional up-regulation of the *EPHA7* [139]	ALL: *EPHA7* hypermethylation [131] T-ALL/lymphoma with 6q deletion: EPHA7 as a tumor suppressor [140] GC B cell NHL: *EPHA7* hypermethylation and repression [94] FL: EPHA7 proposed as a tumor suppressor [141]
EPHA10		ALL: *EPHA10* hypermethylation [131]
EPHB1		pediatric AML: decreased EPHB1 peptide phosphorylation and mRNA expression compared to healthy controls, hypermethylation of the *EPHB1* promoter [142] ALL: *EPHB1* hypermethylation [131]
EPHB2		ALL: *EPHB2* hypermethylation [131]
EPHB3	T-ALL cell lines H9 and E6.1 [73]	ALL: *EPHB3* hypermethylation [131]
EPHB4	Human leukemia/lymphoma cell lines: EPHB4 mRNA expression in 68 of the 70 studied cell lines [143] K562-R cell line (human imatinib-resistant CML cell line): EPHB4 overexpression [144] Philadelphia chromosome-positive ALL cell lines: imatinib resistance mediated by EPHB4 activation [145]	AML: 28% (7/25) of newly diagnosed AML BM samples over-expressed EPHB4 protein [146] CML: EPHB4 mRNA levels in BM cells significantly increased according to clinical stages (increased EPHB4 expression in blast crisis compared to chronic phase) [144] Childhood ALL: prevalent methylation of the *EPHB4* [147] ALL: *EPHB4* hypermethylation [131] PTLDs: EPHB4 expression suppressed in EBV+ PTLDs [148]
EPHB6	T-ALL cell lines H9 and E6.1 [73]	AML: Increased *EPHB6* expression [149] T-cell leukemia/lymphoma: increased EPHB6 expression [150,151] T-ALL: EPHB6 expression confers increased sensitivity to doxorubicin [151] CLL: EPHB6 expression correlated with a high content of ZAP-70 mRNA and a poor prognosis [152]
EphrinA1		ALL: *EFNA1* hypermethylation [131]
EphrinA3		ALL: *EFNA3* hypermethylation [131]
EphrinA4		CLL: High serum levels of a soluble ephrinA4 isoform positively correlated with increasing PB lymphocyte counts and lymphadenopathy [152] CLL: ephrinA4 expression on PB CLL cells inversely correlated with lymphadenopathy [153]
EphrinA5		ALL: *EFNA5* hypermethylation [131]
EphrinB1	Jurkat cells: EphrinB1 enhanced their metastatic potential [154]	Pediatric B precursor ALL: *EFNB1* identified among downregulated DEGs [155] B-ALL: *EFNB1* as an independent prognostic factor for B-ALL; *EFNB1* mRNA levels significantly lower in B-ALL patients compared to controls [155] ALL: *EFNB1* hypermethylation [131]
EphrinB2	Human leukemia/lymphoma cell lines: EphrinB2 mRNA expression in 58 of the 70 studied cell lines [143] Erythroid leukemia-derived cell line: Induction of proliferation and colony formation, upregulation of growth-related gene expression [156]	ALL: *EFNB2* hypermethylation [131]

ALL, acute lymphoblastic leukemia; AML, acute myeloid leukemia; BM, bone marrow; CLL, chronic lymphocytic leukemia; CML, chronic myeloid leukemia; CNVs, copy number variations; EBV, Epstein-Barr virus; FL, follicular lymphoma; GC, germinal center; MDS, myelodysplastic neoplasms; MM, multiple myeloma; NHL, non-Hodgkin lymphoma; PB, peripheral blood; PTLDs, post-transplant lymphoproliferative disorders.

**Table 3 cancers-15-03963-t003:** Therapeutic targeting of the EPH/ephrin axis in hematologic malignancies.

Target	Type of Hematologic Malignancy	Drug	Type of Study	Outcome	Reference
EPHA2	ALL1/AF9 leukemias	EPHA2 mAb (IF7) radiolabeled with Lutetium-177	Preclinical: Mouse model	Survival advantage	[168]
EPHA3	Pre-B-ALL	EPHA3 with α-particle-emitting Bismuth-213 payload	In vitro: EPHA3-expressing leukemic xenografts	Antitumor effect	[169]
	AML, MDS/MPN, MDS, DLBCL, MF	EPHA3 mAb (KB004)	Clinical: phase I	Responses in AML, MF, MDS/MPN, and MDS patients	[170]
EPHA7	Leukemias with ALL1 translocations	ERK inhibitor (indirect EPHA7 signaling inhibition via its downstream target ERK)	In vitro: K562 cells	Induction of apoptotic cell death	[139]
	Lymphomas	EPHA7 delivered by anti-CD20 mAb (Rituximab)	Preclinical: xenografted human lymphomas	Inhibition of ERK and SRC activity; induction of cell death	[141]
EPHB4	Imatinib-resistant CML	Vandetanib	Preclinical: K562 cells	Growth arrest; overcoming of imatinib resistance	[177]
EPHB4	AML cases with high EPHB4 expression	EPHB4 mAb (MAb131)	Preclinical: myeloid leukemia cell lines and human myeloid leukemia xenograft models	Effective against AML in vitro and in vivo	[139]
EphrinB2	T-ALL	Combination of TPD7 and berberine (TAB)	Preclinical: Jurkat cells	Decrease in the levels and phosphorylation of ephrinB2; downregulation of PDZ domain-interacting proteins (syntenin-1 and PICK1); decreased phosphorylation of VEGFR2; inhibition of Rac1; upregulation of PTEN	[175]

ALL, acute lymphoblastic leukemia; AML, acute myeloid leukemia; DLBCL, diffuse large B cell lymphoma; ERK, extracellular signal-regulated kinase; mAb, monoclonal antibody; MDS, myelodysplastic neoplasms; MDS/MPN, myelodysplastic/myeloproliferative neoplasms; MF, myelofibrosis; PDZ, post-synaptic density protein (PSD95), Drosophila disc large tumor suppressor (Dlg1), and zonula occludens-1 protein; PICK1, protein interacting with C Kinase-1; PTEN, phosphatase and tensin homolog; VEGFR2, vascular endothelial growth factor receptor 2.

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
