# Peer review of "EPH/Ephrin Signaling in Normal Hematopoiesis and Hematologic Malignancies: Deciphering Their Intricate Role and Unraveling Possible New Therapeutic Targets"

_cancers, 2023, doi:10.3390/cancers15153963_

Round 1
Reviewer 1 Report
This is a well written comprehensive overview containing important physiological and clinical informations on the role EPHs in hematological malignancies.
Author Response
Response to Reviewer 1
We would like to thank the reviewer for the positive comments. We are glad we have managed to cover both physiological and clinical aspects of EPH/ephrin signaling in normal and malignant hematopoiesis.
Response to Reviewer 2
We would like to thank the reviewer for the positive comments. We are pleased to read that you have appreciated the structure and content of the review.
Response to Reviewer 3
We would like to thank the reviewer for the positive comments.
Response to Reviewer 4
We would like to thank the reviewer for the positive comments. We are glad to read that you have found we adequately covered the addressed topic. Please find below the point-to-point answers to the issues you have mentioned:
- We provide the relevant printing licenses for each of the figures created with biorender.
- After your interesting proposal we revised the title to “EPH/ephrin signaling in normal hematopoiesis and hematologic malignancies: Deciphering its intricate role and unraveling possible new therapeutic targets.” We believe that the revised title summarizes all the topics reviewed in this manuscript more precisely.
- The space has been removed as indicated.
Reviewer 2 Report
In this review article, Stergiou et al. have summarized the most recent findings in EPH/ephrin signalling. In this very comprehensive review, the authors focus on the role of EPH/ephrin signalling in hematologic development and related malignancies. The review is straightforward, clear, and well written. Therefore, I suggest the publication of the manuscript in the current form.
Author Response

(The authors gave the same response as above.)

Reviewer 3 Report
well done
Author Response

(The authors gave the same response as above.)

Reviewer 4 Report
Your manuscript is a well-written review summarizing the impact of EPH/ephrin signaling in the context of normal hematopoiesis as well as hematologic malignancies. Your article will make an important contribution to better understand ephrin signaling in the hematopoietic system. Attached you will find some major/minor issues regarding your manuscript.
Major/Minor Issues:
1. Please provide the printing license for the biorender figure.
2. Please discuss if you like to change your title to “EPH/ephrin signaling in the spotlight: Unraveling its intricate role in hematopoiesis and hematologic malignancies”
3. Line 159: Remove space.
Good English language.
Author Response

(The authors gave the same response as above.)
